# Post-Traumatic Stress Response and Appendicitis in Children—Clinical Usefulness of Selected Biomarkers

**DOI:** 10.3390/biomedicines11071880

**Published:** 2023-07-02

**Authors:** Jarosław Sobczak, Monika Burzyńska, Anna Sikora, Anna Wysocka, Jakub Karawani, Janusz P. Sikora

**Affiliations:** 1Department of Paediatric Emergency Medicine, 2nd Chair of Paediatrics, Central Clinical Hospital, Medical University of Łódź, ul. Sporna 36/50, 91-738 Łódź, Poland; jaroslaw.sobczak@umed.lodz.pl; 2Department of Management and Logistics in Healthcare, Medical University of Łódź, ul. Lindleya 6, 90-131 Łódź, Poland; 3Department of Epidemiology and Biostatistics, Chair of Social and Preventive Medicine, Medical University of Łódź, ul. Żeligowskiego 7/9, 90-752 Łódź, Poland; monika.burzynska@umed.lodz.pl; 4Department of Intensive Care and Anaesthesiology, 2nd Chair of Paediatrics, Central Clinical Hospital, Medical University of Łódź, ul. Sporna 36/50, 91-738 Łódź, Poland; anna.sikora@umed.lodz.pl; 5Department of Paediatric Surgery and Oncology, Chair of Surgical Paediatrics, Central Clinical Hospital, Medical University of Łódź, ul. Sporna 36/50, 91-738 Łódź, Poland; anna.wysocka@umed.lodz.pl; 6Faculty of Medicine, Lazarski University, ul. Świeradowska 43, 02-662 Warsaw, Poland; kubakarawani@gmail.com

**Keywords:** biomarkers, surgical trauma, appendicitis, children

## Abstract

Acute appendicitis is an inflammatory process which is one of the most frequent global causes of surgical interventions in children. The goal of the study was to determine whether acute phase proteins, that is, C-reactive protein (CRP), procalcitonin (PCT) and neutrophil gelatinase-associated lipocalin (NGAL), interleukin 6 (IL-6), transforming growth factor-beta1 (TGF-β1) and cortisol (HC) play a role in the pathomechanism of post-trauma stress response of the organism and to establish the impact of the applied surgical procedure and/or of inflammation on their concentrations. An additional purpose was to establish the clinical usefulness of the studied biomarkers in the diagnostics of appendicitis. CRP concentrations were quantified via the immunoturbidimetric method, while the levels of IL-6 and PCT were assessed using a bead-based multiplexed immunoassay system in a microplate format (Luminex xMAP technology); NGAL, TGF-β1 and cortisol concentrations were determined via the enzyme-linked immunosorbent assay (ELISA) technique. All the investigated biomarkers were assayed twice, i.e., immediately before the surgery and 12–24 h after its completion. Significant increases in CRP, IL-6 and PCT concentrations were found in all children subjected to laparoscopic surgeries (*p* = 0.001, *p* = 0.006, and *p* = 0.009, respectively) and open (classic) surgeries (*p* = 0.001, *p* = 0.016, and *p* = 0.044, respectively) compared to the initial concentrations. The patients undergoing classical surgery moreover presented with significant (*p* = 0.002, and *p* = 0.022, respectively) increases in NGAL and TGF-β1 levels after the procedures. In a group of children undergoing laparoscopic surgery, the appendicitis induced an increase in cortisol concentration, whereas in patients undergoing classical surgery the increase in the levels of this biomarker was caused by the type of performed surgical procedure. Simultaneously assaying the levels of CRP, NGAL and IL-6 (*p* = 0.008, *p* = 0.022, and *p* = 0.000, respectively) may prove useful in clinical practice, enabling the diagnosis of appendicitis in paediatric patients reporting to a hospital with abdominal pains, in addition to data from anamnesis and from clinical or ultrasound examination. The performed study confirms the participation of examined biomarkers in the pathomechanism of post-injury stress reaction of the organism to surgical trauma.

## 1. Introduction

A condition for eliciting a defensive response from the organism to the action of an injury factor is an interruption in the continuity of skin or mucous membranes. This response is comprehensive, and in addition to the activation of the immune system, a neurohormonal response is activated, related, among others, to the secretion of corticotropin, cortisol or catecholamines. An acute infection (bacterial or viral), injury (mechanical/thermal) and ischaemia with subsequent tissue hypoxia results in the release of endogenous inflammation mediators (among others, pro-inflammatory cytokines, and reactive forms of oxygen) and acute phase proteins engaged in humoral innate immune response mechanisms into the plasma [1,2,3]. During controlled inflammatory reaction, pro-inflammatory mediators induce the creation of anti-inflammatory cytokines, the synthesis of which is an expression of the homeostasis of the organism, and the symptoms of inflammation gradually subside. Initially, the inflammation is not generalised, but local, and exudate with a high level of protein is a morphological characteristic of an acute inflammation. However, if the harmful factor is strong enough, and the body is weakened enough to make adaptive anti-inflammatory mechanism insufficient, then the inflammatory process becomes generalised. Clinically, this situation manifests in systemic inflammatory response, which in the literature is referred to as SIRS (systemic inflammatory response syndrome) [4].

Acute appendicitis (AA) is a common clinical example and is one of the most frequent global causes of surgical intervention, with a risk of death estimated at between 6% and 17% [5,6,7]. Approximately, 16.5% of cases of acute appendicitis are complicated by perforation, peritonitis or an abscess [8,9]. The unfavourable development of appendicitis may result in the development of a generalised inflammatory reaction (including, among others, increased production of pro-inflammatory cytokines and acute phase proteins, i.e., CRP, PCT, and NGAL) leading to bacterial sepsis. Particularly susceptible to the generalisation of the inflammatory process are small children with a weakened immune system (e.g., born prematurely, cancer patients). Despite the decrease in mortality resulting from AA in recent years, a delay in correct diagnosis and treatment results in an increase in perioperative complications and also extended duration of hospitalisation [10]. That is why it is so important to diagnose AA early at the stage of differential diagnosis, to allow the appropriate therapeutic treatment to start. Clinical symptoms of appendicitis in children are non-specific. The possibility of overlapping symptoms of other childhood diseases and difficulties in the examination of abdomen during this period of life may delay an appropriate diagnosis. In children below 12 years of age, an initial erroneous diagnosis reaches a range of 28–57% and increases to almost 100% in children below the age of 2 [11,12,13]. Erroneous diagnoses related to AA most frequently include mesenteric lymphadenitis, constipation, gastroenteritis, as well as ovarian cyst torsion and pyelonephritis or intestine cancer (e.g., lymphoma) [14,15,16]. Therefore, diagnosing AA in children is not easy and in all age groups requires a clinical assessment (collection of data from anamnesis and a thorough physical examination), laboratory tests and imaging diagnostics. The use of ultrasound and CT may improve diagnostic accuracy in some patients and reduce the percentage of negative appendectomies. Imaging (radiological) examinations, in particular CT, provide much better diagnostic capabilities, with reported sensitivity and specificity, 94% and 95%, respectively, but are burdened with an increased risk of development of a tumour as a result of irradiation [17]. In the case of laboratory testing, both biochemical and haematological markers are commonly used to increase the precision and reliability in the diagnosis of AA and they provide significant help in differential diagnosis. The use of new markers (e.g., neutrophil lipocalin, and interleukin-6) provides more diagnostic benefits, but incurs significant costs [18].

The goal of the study was to check whether the examined biomarkers, that is, C-reactive protein (CRP), procalcitonin (PCT) and neutrophil gelatinase-associated lipocalin (NGAL), interleukin-6 (IL-6), transforming growth factor-beta1 (TGF-β1) and cortisol (HC) play a role in the pathomechanism of development of post-trauma stress response of the organism, taking into account the possible impact of the used surgical procedure and/or inflammation on their concentrations. Moreover, we wanted to establish the clinical utility of these markers in the diagnostic aspect of the course of appendicitis in children. Finally, the importance of studying those serum inflammatory biomarkers in this clinical entity could be helpful to differentiate the stage of inflammation because most patients are healthy before developing appendicitis, and the length of symptoms before hospitalisation is typically short.

## 2. Materials and Methods

### 2.1. Study Site and Subjects

The study included 40 children of various ages with inflammatory and non-inflammatory diseases (18 patients with appendicitis and 22 with non-inflammatory diseases) hospitalised in the period of May 2017 to April of 2021 at the Paediatric Surgery and Oncology Clinic of the Central Clinical Hospital of the Medical University of Lodz. These patients underwent procedures performed with classical (open) surgical method or with minimally invasive (laparoscopic) method. The study was conducted in accordance with the Declaration of Helsinki, and the protocol RNN/176/17/KF, date 16-th May 2017 (number of the amending resolution KE/620/19, date 9-th April 2019) was approved by the Medical University Ethics Committee, InterDoktorMen project (POWR.03.02.00-00-1027/16-00). The examined patients were in the age range of 6 to 17 years. Each time before a surgery was performed, a written permission for the procedure was obtained from the parents.

The study inclusion criteria were patients aged 5 to 18 years, of female and male sex, admitted to the Paediatric Surgery and Oncology Clinic for surgical procedures, both planned and life-saving. The exclusion criteria included patients with impaired immunity (immune deficits, cancer, autoimmune diseases, and immunosuppressive treatment), patients with chronic inflammatory disease, and a lack of informed consent provided by the examined teenage child/patient or their parents/legal guardians. In our study, we did not use a randomisation of the method of surgical appendectomy, since most of the procedures were performed on an emergency basis, and the preferences of the surgeon decided on the type of procedure. Not all surgeons in our team have mastered the laparoscopic appendectomy technique, which was a limitation of our study.

All patients were administered perioperative intravenous cefazoline and metronidazole. Children diagnosed with gangrenous appendicitis with perforation or limited peritonitis were administered broad-spectrum antibiotic therapy (cefuroxime, metronidazole, and amikacin). The group of patients operated on via the classical (open) method constituted 14 paediatric patients aged 6–17 years (Table 1). In this group, the surgeries performed included appendicitis (n = 2), including a single case of phlegmonous appendicitis, and non-inflammatory diseases (n = 12). The diseases without clinical characteristics of inflammation included inguinal hernia (n = 3), ureteropelvic junction stenosis (n = 2), and also procedures within the area of male genital organs: orchiopexy (n = 2), phimosis (n = 2), hydrocele testis (n = 1), testectomy (n = 1) and testicular prosthesis (n = 1). Laparoscopic surgery was performed on 26 children in the age of 9–17 years (Table 1). These patients were subjected to the following surgical procedures: appendicitis (n = 16), cholecystolithiasis (n = 6), varioceles (n = 2), kidney cysts (n = 1) and explorative laparoscopy procedure (n = 1). Individual analysis of 18 cases with clinical symptoms of appendicitis has demonstrated the morphological characteristics of phlegmonous appendicitis in 8 patients, and of gangrenous appendicitis in 1 child; in the remaining 9 cases catarrhal appendicitis was diagnosed. The clinical symptoms of limited peritonitis were observed in two children, with other symptoms including tenderness in the lower right quadrant of the abdomen, nausea, vomiting, diarrhoea, and increased body temperature (>37.2 °C). The pain which occurred in all children with appendicitis was of a changeable location and occurred most frequently in the lower right quadrant of the abdomen, but also in the whole abdomen, in the right mesogastrium and in the hypogastrium (Table 2). The hospitalisation time of the patients in the study was 2 to 6 days.

Appendicitis was diagnosed based on clinical symptoms and the results of laboratory tests and ultrasound examination. The clinical symptoms that justified the suspicion of appendicitis were the same as those indicated by Alvarado (i.e., migration of the pain, anorexia, nausea, vomiting, right lower quadrant tenderness, rebound tenderness, elevated temperature, leukocytosis, and polymorphonuclear neutrophilia >75%) [19]. The assessed results of laboratory tests have included an increased concentration of CRP (>5 mg/L) and high number of white blood cells (WBC) (6–12 years WBC >13.5 G/l; and >12 years WBC > 10 G/l). Ultrasound criteria of appendicitis included distension of the appendix (>6 mm), thickening of its wall (>2 mm), pain during an attack of the appendicitis when pressed upon by an ultrasound transducer, appendix infiltration and rigidity of the appendix wall when pressed upon by an ultrasound transducer. As an unequivocal diagnosis of appendicitis, the meeting of at least two clinical criteria, one laboratory criterion and two ultrasound criteria was assumed. All excised appendices were sent for routine histopathology evaluation, which was crucial for final diagnosis. The criterion for phlegmonous appendicitis was histopathological evidence of transmural inflammation, and the criterion for advanced appendicitis was evidence of transmural gangrene or perioperative findings of perforation.

### 2.2. Surgical Technique and Qualification for Surgery

All patients were given cefazolin sodium at a dose of 30 mg/kg and metronidazole 10 mg/kg before the operation. Premedication, induction of anaesthesia, endotracheal intubation and maintenance of anaesthesia were performed according to anaesthesia protocols.

Under general anaesthesia, after prior preparation of the operating field, the abdominal cavity was opened in layers via a transverse incision in the right iliac fossa. After muscle dissection, the peritoneum was visualized, cut longitudinally, and then the caecum with the inflamed appendix was exposed. The appendix was amputated and a 2.0 Vicryl and 2.0 Vicryl Z-type suture was placed on its base. The postoperative wound was covered with a sterile dressing.

During the laparoscopic procedure performed under general anaesthesia, three ports were used (a 10 mm trocar was placed under the umbilicus, and two 5 mm trocars—one in the left lateral region and the other in the left inguinal region). After identification of the cecum and the appendix, the mesentery was cut by coagulating the mesenteric vessels with monopolar diathermy, and two ligatures of the Endolup 2.0 type were placed on the base of the appendix, approximately 0.5 cm apart. Then, the appendix between the ligatures was cut and removed in an endobag through the 10 mm trocar site. Finally, the appendix was sent for histopathological examination. The wounds were secured with sterile dressings. Laparoscopic procedures were performed using the Type Image 1S laparoscopic set by Karl Storz. 

The choice of the surgery method was at the discretion of the surgeon and depended on his skills. Nevertheless, even minimally invasive procedures (e.g., laparoscopic) may require conversion to a classic procedure, which is conditioned by difficulties resulting from anatomical anomalies or the course of the disease itself. Qualification for surgery was based on the patient’s clinical condition (e.g., clinical symptoms of an acute abdomen), an increase in the number of neutrophils with a left shift in the smear and elevated concentrations of acute phase proteins (e.g., CRP, PCT, and NGAL). In case of diagnostic doubts, imaging studies were used (e.g., ultrasound, and abdominal CT). In the case of a patient with a history of multiple laparotomies, the preferred method is the classic procedure due to the high risk of damage to internal organs, especially the intestines, during the insertion of trocars. In addition, all patients with respiratory failure and haemodynamically unstable are always qualified for open surgery. The literature on the subject does not clearly indicate the advantage of laparoscopy over the classic approach; however, it indicates the advantage of laparoscopy in several aspects: lower percentage of superficial surgical site infections, shorter hospitalisation time, faster return to daily activity and less intensity of pain in the first days after the procedure.

### 2.3. Factors Predisposing Children to Post-Traumatic Inflammatory Reaction

Performing surgery is traumatic for the child, which causes the post-traumatic inflammatory event. This response of the body will be intensified in the presence of immunosuppression in the operated child (i.e., as a result of the surgery performed, and concomitant diseases, e.g., autoimmune diseases; and malignancies, or medications, e.g., glucocorticosteroids), which may additionally result in the generalisation of the inflammatory process, i.e., the development of sepsis complicating the performed surgery. Fortunately, postoperative immune complications, such as poor wound healing and infection, are rare in patients with a properly functioning immune system. In addition, children with concomitant infection, with a long period of time between the clinical manifestation of appendicitis and admission to the hospital, and those with an increased severity of the disease process (e.g., pyogenic or gangrenous appendicitis with accompanying peritonitis), will present an increased post-traumatic inflammatory reaction. 

It should also be mentioned that not only surgical trauma, caused by tissue damage, influences the post-traumatic inflammatory event, but also the drugs and techniques used during anaesthesia. The intensity of the cytokine response is directly related to the size and scope of surgical wound, which depends mainly on the choice of the surgical method, i.e., a slight increase in the concentration of proinflammatory cytokines has been observed during procedures performed with the minimally invasive method (e.g., laparoscopy, and thoracoscopy), increasing up to high levels in extensive open surgery [18]. It is believed that regional anaesthesia suppresses the immune system to a smaller extent, in comparison to general anaesthesia. Intravenous anaesthetics have anti-inflammatory properties that may be beneficial for most septic patients, due to the possibility of protection against organ dysfunction after endotoxemia as a result of the action of pro-inflammatory cytokines and oxygen free radicals.

### 2.4. Methods and Collection of Specimens

Blood for tests was collected twice from a venous catheter while taking other tests routinely performed for surgery. Determination of the concentrations of individual biomarkers (80 samples in total) was performed twice in each patient, i.e., the first sample was collected just before the surgery, and the second one within the first 12–24 h after its completion. First, whole blood samples (1.5 mL) were collected from the studied children by venepuncture into Microtainer Serum Separator tubes, and then, after clotting, the serum was centrifuged immediately at 1000× *g* for 10 min, after which it has been frozen and stored at a temperature of −80 °C until assayed for individual determinations. Evaluation of all the immunological markers was carried out in the Research Laboratory CoreLab, Central Clinical Hospital, Medical University of Lodz.

#### 2.4.1. C-Reactive Protein Assay

CPR was quantified with the Alinity c CRP Vario test, using the immunoturbidimetric method, in human serum at variable measuring ranges (CRP16 and CRP48) on the Alinity c analyser. Alinity c CRP Vario is an immunochemical test that uses latex particles. When an antigen–antibody reaction occurs between the CRP protein present in the test sample and the anti-CRP antibody adsorbed on the latex particles, an agglutination process takes place. The agglutination process is detected as a change in absorbance (572 nm), the rate of change being proportional to the amount of CRP present in the test sample.

#### 2.4.2. Interleukin-6 Assay

The level of IL-6 in the serum was determined using a kit based on the Luminex xMAP technology (cat. no. FCSTM09, R&D Systems, Minneapolis, MN, USA). Magnetic beads pre-coated with antibodies specific for IL-6 were added to the wells with standards and samples. During the incubation step, the analyte from the sample was captured by the beads. After a washing step, biotinylated detection antibodies were added. Unbound antibodies were washed off and streptavidin–PE conjugate was added to the wells to detect biotinylated antibodies on the surface of each bead. In the next washing step, unbound streptavidin-PE was removed and the magnetic beads were resuspended in buffer. The amount of bound analyte proportional to the fluorescent signal was then quantified. The fluorescence was read with a Luminex^®^ MAGPIX^®^ analyser (Luminex, Austin, TX, USA). All results were analysed with Belysa 1.1.0 Software and the protein concentration per ml was determined by interpolation from the standard curve referring to the five-parameter logistic (5-PL) curve.

#### 2.4.3. Neutrophil Gelatinase-Associated Lipocalin Assay

Serum NGAL level was determined with an ELISA kit based on the sandwich principle according to the manufacturer’s instructions (cat. no. orb219505, Biorbyt, Cambridge, UK). Standards and samples were added into wells pre-coated with an antibody specific for human NGAL. The antigen present in the sample was bound by the immobilized antibody and any unbound substances were washed off. The biotinylated anti-human NGAL antibody, HRP-conjugated streptavidin and TMB substrate solution were then sequentially added to the wells, with washing and incubation steps in between. The colour intensity was proportional to the amount of NGAL bound in the initial stage.

#### 2.4.4. Transforming Growth Factor-Beta1 Assay

Serum TGF-β1 level was determined with an ELISA kit based on the sandwich principle according to the manufacturer’s instructions (cat. no. EIA-1864, DRG, Marburg, Germany). Prior to testing, standards and samples were diluted in an assay buffer, acidified with HCl and neutralized with a neutralization buffer. The neutralized standards and samples were added to wells coated with polyclonal antibody. After an overnight incubation, unbound sample material was removed by washing. Subsequently, a monoclonal mouse anti-TGF-β1 antibody, a biotinylated anti-mouse IgG antibody, and a streptavidin-HRP enzyme complex were sequentially incubated with washing and incubation steps in between, forming an immune enzyme sandwich complex. After the incubation step, the unbound conjugate was washed off and a TMB substrate solution was added to the wells. The enzymatic reaction was stopped with H_2_SO_4_ and the colour intensity (absorbance) proportional to the concentration of TGF-β1 in patient sample was measured.

#### 2.4.5. Procalcitonin Assay

Serum PCT level was measured with a kit based on the Luminex xMAP technology (cat. no. LXSAHM, R&D Systems, Minneapolis, MN, USA). Standards, samples and magnetic beads coated with procalcitonin-specific antibodies were added to the wells of a 96-well plate. During the incubation step the analyte from the sample was captured by antibodies. In the next step, any unbound substances were washed away and to detect biotinylated antibodies on the surface of each bead, streptavidin–phycoerythrin conjugate was added to the wells. Any unconjugated and free dyes were removed with final washes and the beads were resuspended in buffer. The fluorescence was read with a Luminex^®^ MAGPIX^®^ analyser (Luminex, Austin, TX, USA) to determine the magnitude of the PE-derived signal, which was directly proportional to the amount of bound analyte. All results were analysed with Belysa 1.1.0 Software and the protein concentration per ml was determined by interpolation from the standard curve referring to the five-parameter logistic (5-PL) curve. 

#### 2.4.6. Cortisol Assay

Serum cortisol level was determined with an ELISA kit based on competitive binding principle according to the manufacturer’s instructions (cat. no. DEH3388, Demeditec, Kiel, Germany). Standards, samples and horseradish peroxidase-labelled cortisol were added to microtiter wells coated with an anti-cortisol antibody. An unknown amount of cortisol from the sample competed with a cortisol–HRP conjugate for binding to the coated antibody. After the incubation step, any unbound substances were washed off. The amount of bound peroxidase conjugate was inversely proportional to the concentration of cortisol in the sample, therefore, after the addition of the substrate solution, the colour intensity was also inversely proportional to the cortisol concentration in the sample.

A Stat-Matic Plate Washer II (Sigma-Aldrich, Taufkirchen, Germany) was used for the washing steps in the ELISA assays and the absorbance was read via Multifunctional Microplate Reader VICTORTM X4 (Perkin Elmer, Shelton, CT, USA). All ELISA results were analysed with WorkOut 2.5 Software and the mean concentration of protein per ml was determined by interpolation from the standard curve referring to the four-parameter logistic (4-PL) curve.

### 2.5. Statistical Analysis

The data was coded and entered into Microsoft Office Excel and STATISTICA version 13.1. The most important computational methods in the field of descriptive statistics were used in the statistical analysis, including the measures of the distribution position for measurable features: arithmetic mean to calculate the average level of the analysed statistical feature in the population, standard deviation for the assessment of dispersion for measurable features, and median to calculate the median value of the feature for the study group in a situation where the distribution of the feature deviated from the normal distribution. Moreover, the structure indexes interpreted in fractions (for the number <100) were calculated to assess the ratio of a part of a given statistical population distinguished by a specific level/variant of a feature to the entire studied population. Moreover, on the basis of selected methods in the field of analytical statistics, the relationships between selected statistical features were calculated and the significance of differences in the selected variables was assessed. For this purpose, Student’s *t*-test and the Mann–Whitney U test, as well as the Wilcoxon pairs test and the difference test were used to compare the two groups in terms of a quantitative variable. The Shapiro–Wilk test was used to assess the normality of the distribution. The statistical analysis also assessed the presence of correlation between the levels of biomarkers. The quantitative nature of the correlated variables made it possible to use the rectilinear correlation coefficient (r) for this purpose. The research hypotheses were verified on the basis of the significance level *p* ≤ 0.05.

## 3. Results

By assessing the CRP levels in all children, i.e., those without inflammation and with inflammation, operated on with the open method and with laparoscopy, significant (*p* = 0.001, and *p* = 0.000, respectively) increases in CRP concentrations after the surgery were demonstrated in comparison to the initial values (Me of 0.35 vs. 3.40 and Me of 2.30 vs. 31.30, respectively). A similar tendency was observed in the group of children with inflammation operated on via the laparoscopic method (*p* = 0.013, and Me of 3.85 vs. 78.90). The same trend in CRP levels occurred in patients without inflammation, operated on both using the open method (*p* = 0.003, and Me of 0.3000 vs. 3.2000) and the endoscopic method (*p* = 0.005, and Me of 1.30 vs. 8.15) (Table 3 and Table 4).

By analysing the IL-6 concentrations in all children subjected to the procedures, a significant (*p* = 0.000) increase in its levels after the procedure compared to the concentrations before the procedure was observed (Me of 3.12 vs. 9.00). A more thorough assessment of the levels of this biomarker conducted over the same periods of time in all the examined patients have demonstrated a similar increasing tendency in both patients undergoing classical surgery (*p* = 0.016, and Me of 2.65 vs. 7.84) and those subjected to laparoscopic procedures (*p* = 0.006, and Me of 4.74 vs. 12.82) (Table 3 and Table 4). In turn, the analysed initial concentrations of IL-6 were significantly (*p* = 0.018, and *p* = 0.036, respectively) higher in children with symptoms of appendicitis compared to the group with no inflammation before the surgery, performed with both laparoscopic and open method (Me of 6.99 vs. 2.66 and 14.79 vs. 2.55, respectively) (Table 3). No statistical significance in IL-6 levels was observed when comparing the concentrations of this biomarker in analogous groups of patients after laparoscopic and open surgeries.

Moreover, children with inflammation and those without, undergoing classical surgery, presented a significant (*p* = 0.002, and Me of 33.99 vs. 39.28) increase in NGAL concentrations after the procedure was conducted. A similar tendency in the levels of this marker was observed in the group of children without inflammation operated on using the open method (*p* = 0.004, and Me of 31.45 vs. 38.02) (Table 3 and Table 4).

Assessing the TGF-β1 concentrations in all children with and without inflammation a significant (*p* = 0.022, and Me of 0.45 vs. 0.82) increase in its level was demonstrated after procedures performed with the open method. Similarly, increased concentrations of this marker were observed in patients with inflammation undergoing laparoscopic surgery (*p* = 0.041, and Me of 0.49 vs. 0.69) (Table 3 and Table 4). 

A significant increase (*p* = 0.044, and *p* = 0.009, respectively) in PCT concentrations was demonstrated in all children with and without inflammation after surgeries were performed using open and laparoscopic method, and arithmetic means amounted to 514.08 vs. 678.94 and 521.41 vs. 820.07, respectively. Similarly, patients with inflammation undergoing laparoscopic surgery presented significantly (*p* = 0.007) higher levels of this marker after the procedure compared to initial values (Me of 514.05 vs. 563.18) (Table 3 and Table 4). 

Comparing cortisol levels in the studied groups of patients, no statistically significant changes in the concentrations of this marker were demonstrated after both the surgical procedures. 

Analysing the difference in the levels of individual biomarkers between four examined groups of children (group 1—laparoscopic non-inflammatory, group 2—laparoscopic inflammatory, group 3—open non-inflammatory, and group 4—open inflammatory), an attempt was made to establish whether the type of conducted surgical procedure, and also the presence or lack of inflammation had an impact on the concentrations of examined biomarkers. This analysis has demonstrated statistically significant differences of cortisol concentrations (*p* = 0.047, chi-square test), that is, patients with appendicitis operated on with the laparoscopic method have presented a higher level of cortisol compared to those without inflammation (Me of 16 vs. 10), which indicates that in these children the inflammation has caused an increase in cortisol concentration. On the other hand, children operated on with the open (classic) method have presented a higher concentration of cortisol in non-inflammation group compared to those with appendicitis (Me of 12 vs. 2), which means that the type of surgical procedure conducted has induced an increase in the level of this marker.

When comparing the levels of individual biomarkers before the surgery between all patients with appendicitis (n = 18) and those without (n = 22) using a Mann–Whitney test, significantly higher concentrations of both CRP (*p* = 0.008, and Me of 0.45 vs. 4.30) (Figure 1) and NGAL (*p* = 0.022, and Me of 33.99 vs. 46.15) (Figure 2) have been demonstrated. Similarly, significantly higher levels of IL-6 were observed in analogous groups of patients (*p* = 0.000, and Me of 2.59 vs. 7.82, respectively) (Figure 3) by analysing the initial (preoperative) values of this marker.

By looking for a connection between postoperative concentrations of the tested biomarkers after laparoscopic procedures and open method procedures, statistically significant correlations were observed between CRP and NGAL (r = 0.3674; and *p* = 0.020) as well as between CRP and PCT (r = 0.6086; and *p* = 0.000). Similarly, a positive correlation was demonstrated between NGAL and TGF-β1 (r = 0.3355; and *p* = 0.034). By analysing the dependencies between IL-6 levels after surgical procedures and the rest of the markers, significant correlations of this cytokine with CRP, NGAL and PCT were revealed (r = 0.5082 *p* = 0.001; r = 0.3820 *p* = 0.015; and r = 0.3323 *p* = 0.036, respectively) (Table 5). An analysis of postoperative correlations between CRP and TGF-β1, cortisol and between PCT and TGF-β1, cortisol, as well as between NGAL and PCT, cortisol showed that they were not statistically significant.

## 4. Discussion

The neuro-hormonal response of the organism activated simultaneously by the immune and inflammatory response to a triggering stress factor (infection, injury, and ischaemia) occurs through the activation of the hypothalamic–pituitary–adrenal axis (HPA) and of the sympathetic–adrenal system. The hypothalamus is a centre which integrates and regulates the activity of these systems. We first observe an increased secretion of catecholamines (adrenaline and noradrenaline) which reflects the activated sympathetic–adrenal system, with clinical consequences of increased action of the cardiovascular system, distension of skeletal muscle vessels and constriction of skin and gastrointestinal vessels, which increases blood flow to brain and muscles. Simultaneously, the availability of energy substrates increases and the processes of glycogenesis and gluconeogenesis in the liver intensify, lipolysis is stimulated and the secretion of insulin is inhibited. Further, post-traumatic stress reaction activates the HPA axis, with effects visible for a longer period. The released cortisol will increase gluconeogenesis and proteolysis and will have an anti-inflammatory effect as well as increase insulin resistance of cells. In the paracrine neurohormonal response to a stress factor, the prevailing opinion is that many cytokines (in particular IL-1, IL-6, and TNF-α) may activate the HPA axis and at least partially intermediate the answer to various stimuli which cause the feeling of pain. It assumed that pro-inflammatory cytokines, acting both on the level of hypothalamus, pituitary gland and adrenal glands stimulate, among others, the synthesis of glucocorticosteroids by the adrenal cortex. Therefore, this may foster the intensification of immunosuppression, with clinical manifestation of, e.g., development of sepsis which complicates a general injury [20].

Acute phase proteins are considered to be both objective markers of post-traumatic inflammatory response and mediators of immune response, similar to cytokines. They are comprehensively used in the monitoring of patients with SIRS and are used in particular when microbiological diagnostics are hindered or ineffective, and when the primary disease or treatment (immunosuppression) is masking clinical and laboratory symptoms. One of the acute phase proteins which participate in the innate humoral immunity is the C-reactive protein, and its increase is induced by pro-inflammatory cytokines (e.g., IL-1, IL-6, and TNF-α) released by phagocytes during the innate, immediate immune response [21]. CRP appears then as a defensive response of our body, and its main function is to stimulate the response of the immune system, consisting of, among others, deactivation of inflammatory factors and direct stimulation of immune system cells [22]. The CRP level is only slightly modified by hormones, anti-inflammatory drugs and other biological substances, which is why it is a reliable marker used to monitor SIRS in the organism. Jeschke et al. in their analysis of a very large group of children with extensive burns conclude that CRP may be used as a biomarker of homeostasis in these patients, but is not a prognosticative indicator of severe infections and/or sepsis [23]. However, this protein has been reported to be 33% to 95% specific for appendicitis in children with acute abdominal pain. Limited studies, including subjects at the time of surgery, suggest that CRP may be more sensitive (83% to >90%) in detecting appendiceal perforation and abscess formation, conditions more commonly found in children [24]. However, Swedish authors have reported that among new inflammatory markers, serum amyloid A (SAA), matrix metalloproteinase (MMP-9), and myeloperoxidase (MPO) were the strongest discriminators of all cases of appendicitis [25]. On the other hand, Naqvi et al. have observed significantly higher plasmatic levels of IL-6, CRP, monocyte chemoattractant protein-1 (MCP-1), PCT and SAA in children with acute appendicitis compared to patients with non-appendicitis abdominal pain [1]. Our research has confirmed significantly higher CRP values in children with clinical symptoms of acute appendicitis at the moment of hospital admission (Figure 1). By performing a detailed analysis of CRP, we have demonstrated a significant increase in the concentrations of this marker over a specific time period among the examined sub-groups of patients, which indicates that the increase in CRP is induced regardless of the used surgical procedure or the presence of inflammation (Table 3 and Table 4). Our observations match the reports from other authors. Matuszczak et al. in an analysis have also demonstrated a significant increase in CRP after both laparoscopic and open surgery procedures [26]. Ping Li et al. have presented a significantly higher level of CRP and leukocytes in the 5th day after the surgery in a group of children undergoing classic surgery, which may indicate the impact of the type of procedure on the concentration of inflammatory markers [27]. It is commonly assumed that post-traumatic surgical stress induces the production of pro- and anti-inflammatory cytokines and an increase in the level of leukocytes and CRP. This is confirmed in the reports of Chinese authors, who have analysed the post-traumatic response after cholecystectomy with various methods and have suggested a lower stress response in case of laparoscopic surgeries [28].

Cytokines are a group of proteins which include, among others, interleukins which are produced from activated leukocytes, fibroblasts and endothelial cells as an early response to tissue damage, playing the main role in inflammation induced by an infection or by trauma/surgery. The IL-6 is a multi-purpose marker of inflammation produced mainly by monocytes and macrophages; it is considered an early and sensitive, although non-specific marker of inflammation which stimulates the expression of acute-phase proteins in the liver, in particular CRP. The synthesis of IL-6 is stimulated by the action of other pro-inflammatory cytokines (e.g., TNF-α, and IL-1β); its concentration starts to increase approx. 30–60 min from the beginning of the surgery, reflecting the scale of tissue damage. It is more important in minimally invasive procedures (e.g., laparoscopy), whereas high IL-6 concentrations were observed after extensive surgical procedures, reaching a maximum after 24 h and remaining elevated for 48–72 h after the procedure [29]. Due to the fact that during sepsis IL-6 levels usually reach high values, the literature data indicates that they may also be a useful marker in the diagnostics of bacterial infections [30]. Although the mechanism of this increase in acute appendicitis was not precisely established, the recruitment of activated neutrophiles and macrophages in the focus of infection (among others, as a result of structural damage to the appendix) allows the presumption that the increase in metabolic activity of these cells may result in increased levels of IL-6 [31]. On the other hand, high concentrations of this mediator may nevertheless limit the development of inflammation, since IL-6 can block the synthesis of pro-inflammatory cytokines through a feedback inhibition mechanism. The significantly increased initial levels of IL-6 in patients with inflammation observed in our own research may prove a clinically useful diagnostic marker of appendicitis (Figure 3). The significant increases in post-operative concentrations of IL-6 both in patients undergoing classical surgery and those subjected to laparoscopic procedures indicate that the production of this marker increased regardless of the type of surgical procedure (Table 3 and Table 4). Tissue injury at a cellular level resulting from hypoxia or changes to pH caused by the procedure may cause an increase in the IL-6 level already at an early stage. It is therefore considered that the concentration of this biomarker in plasma is an early, sensitive and reliable indicator of tissue injury. The significant increases in IL-6 levels observed after surgical procedures confirm what has been reported in the literature. Chinese authors have described a significant increase in IL-6 after appendectomy procedures in a group of children operated on, regardless of the chosen method, indicating the additional advantages of decreased injury and lower inflammation after laparoscopic procedures [32]. On the other hand, Turkish researches have not demonstrated an increase in IL-6 after laparoscopic procedures, however, after conventional appendectomy they did observe significantly higher levels of this marker compared to preoperative concentrations [33]. In our studies, we have demonstrated significantly higher preoperative levels of IL-6 in children with confirmed appendicitis compared to patients without inflammation, which is in accordance with the reports of other authors and may be predictive for diagnosis [34,35]. Although Spanish scientists indicate that the diagnostic performance of IL-6 alone for the diagnosis of acute appendicitis is limited but they also present the evidence regarding its capacity to discern between complicated and uncomplicated appendicitis in children [31,36]. On the other hand, Kakar et al. have presented in children with appendicitis higher concentrations of both IL-6 and NGAL as markers with clinical utility in the diagnostics of this inflammation, simultaneously emphasising the importance of IL-6 in the assessment of the inflammatory process severity [18].

Lipocalin-2 (LCN-2) is a glycoprotein also known in the literature under the name of siderocalin or neutrophil gelatinase-associated lipocalin (NGAL), which is secreted from the inflammatory cells and tissues as a result of neutrophil activation. NGAL has bacteriostatic properties, which play an important role in the destruction of iron during antibacterial innate immune response. In addition to its important role in the innate response, this property provides it with a protective role in case of injuries, systemic inflammations and various other types of cellular stress; there are attempts to use this inflammatory biomarker in kidney and liver disorders, tumours and inflammatory diseases of the colon [37,38]. Our research has indicated that the levels of NGAL have increased significantly in children after open surgery, suggesting that the higher tissue trauma is related to a higher production of this biomarker than in the case of minimally invasive procedures (Table 3 and Table 4). Therefore, one may conjecture that classical surgery with opening of the abdomen causes a stronger stress reaction of the body to surgical trauma. Similarly, Venge has described that NGAL concentrations are slightly elevated due to major surgical trauma, but the elevations in relative terms are minor as compared to neutrophil counts and CRP [39]. Moreover, our examinations have demonstrated significantly higher initial NGAL concentrations in children with inflammation compared to those that do not present clinical symptoms of inflammation, suggesting the clinical utility of this marker in the diagnostics of appendicitis (Figure 2). Similar conclusions have been drawn based on the results obtained by other authors, which have emphasised the assaying of this mediator as a promising novel biomarker useful in clinical practice in the differential diagnostics of abdomen pains [18,40]. In turn, one of the prospective validation studies conducted with a large sample size confirms that the diagnostic yield of NGAL in the context of paediatric acute appendicitis is only moderate, and therefore, it should not be used as a unique diagnostic tool [41]. On the other hand, Swedish authors have reported that the diagnostic accuracy of NGAL determination additionally exhibited sensitivities and specificities of >90% in most infectious diseases and was clearly superior to current tests, such as the number of neutrophiles in the blood, the C-reactive protein, and procalcitonin or expression of CD64 on neutrophiles in the blood [39]. In our study, we showed the important role of NGAL as a new inflammatory biomarker, which, when measured simultaneously with other markers of inflammation (i.e., CRP, and IL-6), turned out to be helpful (apart from clinical examination and anamnestic data and those obtained from imaging studies) in the diagnosis of appendicitis. Thus, the simultaneous determination of the above-mentioned biomarkers and finding their increased concentration in patients reporting to a paediatric surgeon due to abdominal pain may confirm the presence of inflammation, and in the case of coexisting clinical symptoms suggestive of appendicitis, determine the diagnosis.

TGF-β1 is a cytokine produced by multiple lineages of leukocytes, stromal and epithelial cells, for which many cells of the body exhibit expression of receptors. It is a peptide with two functions: strong pro-inflammatory and immunosuppressive properties; it controls proliferation and differentiation of most types of cells. Immediate secretion of TGF-β1 after the action of an injury exerts strong indirect and direct general pro-inflammatory and immunosuppressive action [42]. In our studies, we have observed a significant increase in TGF-β1 concentration in patients operated on with classic method compared to children undergoing laparoscopic surgery, indicating the leading contribution of tissue in post-traumatic stress response (Table 3 and Table 4). Taha et al. point out that TGF-β1 is the main modulator of the healing process after tissue injuries, and its release is stopped after this process is ended by a feedback mechanism [43]. Other authors have reported that this biomarker remains elevated in the serum long after the formation of scar tissue. The results of our study also indicate the significant increase in TGF-β1 levels after laparoscopic appendectomies (Table 3 and Table 4). On the other hand, Brokelman et al. did not demonstrate significant differences in the concentrations of TGF-β1 in peritoneal biopsies collected in the course of laparoscopic cholecystectomy [44].

PCT is the precursor of calcitonin (CT) produced in the thyroid gland by C cells, however, its exact function in the cytokine cascade and its role in the pathomechanism of inflammatory reaction has not been fully understood yet. In clinical practice, it is used in the diagnosing and differentiating of severe infections and septic complications. Monitoring its level enables revealing early septic complications, in particular in patients with SIRS and multi-organ failure. In these severe medical conditions, PCT is important for prognosis and may be used to assess the effectiveness of antibiotic therapy [45]. The studies conducted so far indicate a selective increase in PCT concentrations in bacterial infections. Therefore, there is a relationship between the level of the examined marker in the blood and the intensification of the patient’s inflammation, which may help in the differential diagnosis of acute appendicitis [46]. Our research has indicated that the PCT concentrations have significantly increased over time in all children, both operated on with the open method and laparoscopic method, which indicates that the synthesis of this biomarker did not depend on the used surgical procedure. Moreover, our results also indicate a significant increase in the level of this marker after laparoscopic appendectomies, suggesting the impact of the inflammatory process on the production of PCT (Table 3 and Table 4). Undoubtedly, bacterial endotoxins, in addition to the proinflammatory cytokines, promote the release of PCT as an early marker of systemic bacterial infection. Haghi et al. have moreover suggested that concurrent assaying of PCT and IL-6 in patients with acute appendicitis has a diagnostic value with sensitivity and specificity of 95% and 55%, respectively [34].

In our study, cortisol has been investigated because of its recognition as an important marker in the pathogenesis of post-traumatic stress response. The prevailing view is that pro-inflammatory cytokines (especially IL-1, IL-6, and TNF-α) can activate HPA axis with subsequent stimulation of glucocorticoid secretion by the adrenal cortex and mediate, at least in part, the response to various nociceptive (pain sensation) stimuli. On the other hand, glucocorticoids demonstrate immunosuppressive activity, inhibiting the secretion of proinflammatory cytokines and limiting the migration of immunocompetent cells to the inflammatory focus; however, they increase the production of anti-inflammatory cytokines. Therefore, we were interested in whether there would be any relationship between cortisol and the cytokines studied. Prete et al. have performed a systematic review and meta-analysis of the stress response of an organism in the form of production of cortisol induced by the surgical procedure, establishing that the cortisol response to the surgical stress significantly differs in patients subjected to minimally invasive procedures (grade I) compared to patients subjected to moderately and highly invasive procedures (grade II and III). In accordance with the analysis by these authors, the level of cortisol on the day of surgery has increased 2-fold (grade I), 4-fold (grade II) and 3.5-fold (grade III) compared to the reference population of healthy people [47]. It may thus be inferred that the response of the organism to surgical trauma seems more intense in case of a more severe injury to the tissue. Our research confirms this hypothesis. Our observations, assessing the impact of the performed surgical procedure and the presence or absence of inflammation on the stress response of the organism have indicated that in the group of children undergoing laparoscopic surgery, the increase in cortisol concentration was induced by appendicitis, whereas in the group of patients operated on with the open method, the increased level of this biomarker was generated by the type of performed surgical procedure. Interesting observations were made by Russian scientists who attempted to justify the operation in children with chronic appendicitis by pointing that autoimmune and vascular mechanisms of appendix damage in children with chronic recurrent lower quadrant pain and laparoscopic appendectomy helped to eliminate abdominal pain in the majority of these patients [48].

Significant correlations between postoperative concentrations of CRP and NGAL and of CRP and PCT (*p* = 0.020, and *p* = 0.000, respectively) demonstrated in our studies may suggest the participation of C-reactive protein in the production of neutrophile lipocalin and procalcitonin, induced by the performed surgical procedures. Similarly, the significant correlations we have observed between IL-6 and CRP, PCT, and NGAL (*p* = 0.001, *p* = 0.036, and *p* = 0.015, respectively) may confirm the literature data on the stimulating impact of IL-6 on the synthesis of acute phase proteins (Table 5). CRP, and PCT are examples of these proteins, the synthesis of which in hepatocytes is stimulated by the action of pro-inflammatory cytokines (among others, IL-6, IL-1β, and TNF-α) in inflammation or tissue damage; they are used globally as a marker of inflammations and infections [49]. However, the main factor contributing to the increasing concentrations of NGAL in the serum/plasma of patients with inflammation is the population of activated neutrophiles, the presence of which in the foci of infection may be induced, among others, by high concentrations of IL-6 [38].

To recapitulate, the ability to recognise and prognosticate the course of a systemic inflammatory reaction in the examined patients as well as the possible occurrence of various complications is very important for a clinician. It should be emphasised that assaying biochemical markers only may prove insufficient in the final effect. That is why they should be always considered in combination with clinical data obtained through an anamnesis, physical examination or conducted imaging studies. However, due to the deficiencies (including low specificity and sensitivity) of the inflammatory reaction biochemical markers used currently, there is ongoing global research directed on the discovery of new biomarkers of the systemic post-traumatic stress reaction of the body.

### Study Limitations

Lack of influence by the research team on the selection of the surgical procedure resulting from the diversity of skills and experience of the operators performing the procedures. In addition, the study assumed a specific period of its duration, which was caused by the financial and logistical capabilities of the research team.

## 5. Conclusions

Our research confirms the participation of CRP, IL-6, NGAL, TGF-β1, PCT and cortisol in the pathomechanism of the development of post-traumatic stress reaction of the organism to surgical trauma. Appendicitis has induced an increase in cortisol level in the group of children subjected to minimally invasive (laparoscopic) procedures, whereas in patients subjected to invasive (open method) procedures, the increase in the levels of this biomarker was caused by the type of conducted surgical procedure. Our research indicates that parallel assaying of the level of CRP, NGAL and IL-6 increases the probability of diagnosing of appendicitis in paediatric patients presenting at an emergency department with clinical symptoms which suggest the presence of this disease.

## Figures and Tables

**Figure 1 biomedicines-11-01880-f001:**
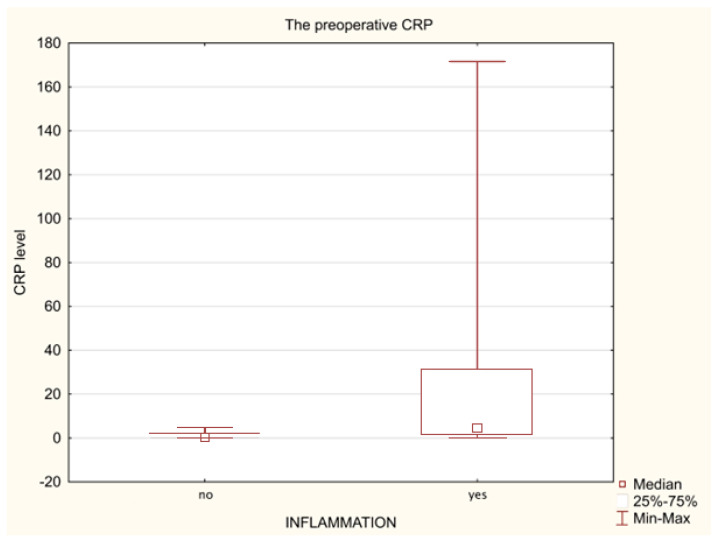
The preoperative CRP levels in the non-inflammatory patients (n = 22) and appendicitis patients (n = 18).

**Figure 2 biomedicines-11-01880-f002:**
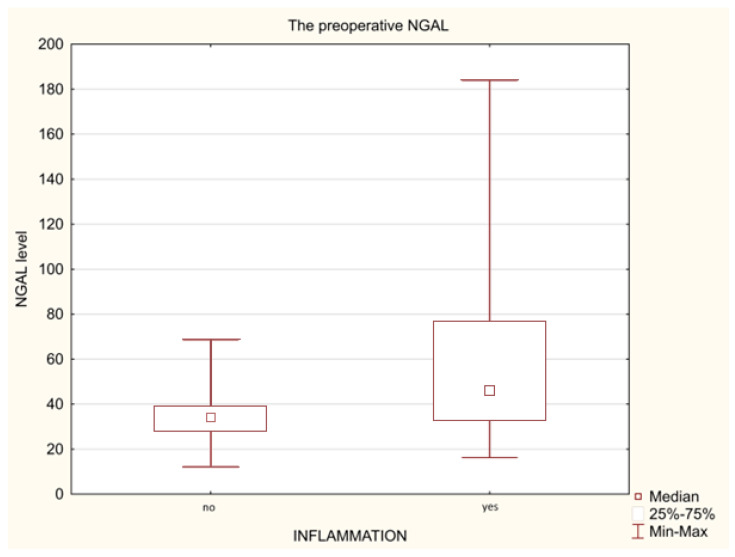
The preoperative NGAL levels in the non-inflammatory patients (n = 22) and appendicitis patients (n = 18).

**Figure 3 biomedicines-11-01880-f003:**
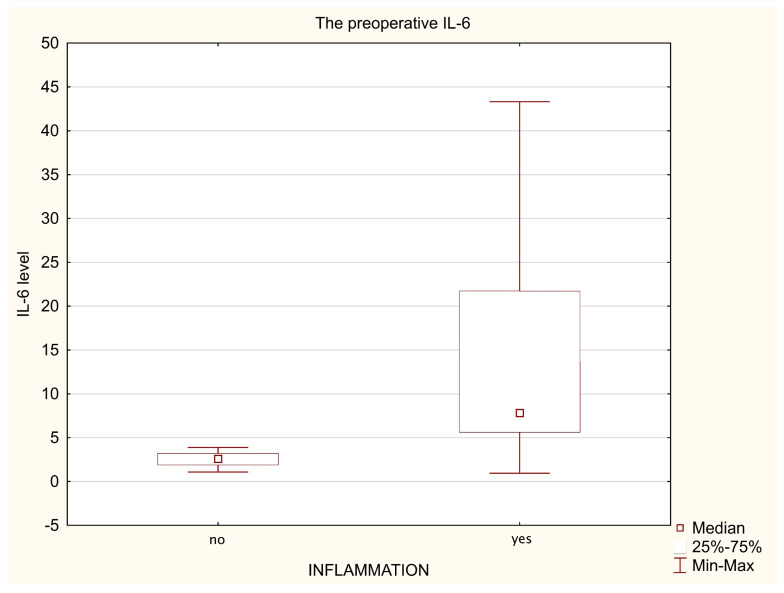
The preoperative IL-6 levels in the non-inflammatory patients (n = 22) and appendicitis patients (n = 18).

**Table 1 biomedicines-11-01880-t001:** The characteristics of the studied groups.

Characteristics		Total n = 40			
		Open Surgery n = 14		Laparoscopy n = 26	
		Inflammatory	Non-Inflammatory	Inflammatory	Non-Inflammatory
		(n = 2)	(n = 12)	(n = 16)	(n = 10)
Age	Minimum	11	6	9	12
	Maximum	17	17	17	17
	Mean	14	11.9	14.8	15.5
	Standard deviation	4.2	4.3	2.3	1.9
Sex N (fractions)	Male	2(1)	11(0.92)	10(0.62)	3(0.3)
	Female	0	1(0.08)	6(0.38)	7(0.7)

**Table 2 biomedicines-11-01880-t002:** The pathology and clinical characteristics of the children with appendicitis.

Patients	n = 18	
Sex	Male, n = 12; Female, n = 6	
Hospitalization time	2–6 days	
Appendix pathology	Catarrhal	9 pts
	Phlegmonous	8 pts
	Gangrenous	1 pt
Body temperature (≥37.2 °C)	6 pts	
Lower right quadrant tenderness	16 pts	
Peritoneal symptoms	2 pts	
Pain localization	Lower right quadrant	12 pts
	The whole abdomen	2 pts
	Right mesogastrium	2 pts
	Hypogastrium	2 pts
Nausea	5 pts	
Vomiting	6 pts	
Diarrhoea	2 pts	
Ultrasound confirmation of appendicitis	18 pts	

Note: pt, patient; pts, patients.

**Table 3 biomedicines-11-01880-t003:** The descriptive characteristics of the studied biomarkers preoperatively.

Characteristics	Laparoscopy			Open Surgery		
	Total n = 26	No Inflammation n = 10	Inflammation n = 16	Total n = 14	No Inflammation n = 12	Inflammation n = 2
C-reactive protein (mg/L)						
Mean	19.8077	1.6300	31.1688	1.2429	0.6250	4.9500
Median	2.3000 *	1.3000 *	3.8500 *	0.3500 *	0.3000 *	4.9500
Standard deviation	45.4710	1.4893	55.5582	1.8232	0.9545	1.0607
Interquartile range	4.200	2.2000	37.0000	0.5000	0.2500	1.5000
Minimum	0.1000	0.1000	0.1000	0.1000	0.1000	4.2000
Maximum	171.600	4.500	171.600	5.700	3.600	5.700
Interleukin-6 (pg/mL)						
Mean	11.3638	3.2110	16.4594	4.4393	2.7133	14.7950
Median	4.7400 *	2.6650 *	6.9950 *	2.6500 *	2.5550 *	14.7950 *
Standard deviation	18.1016	2.0064	21.7185	5.3241	1.4077	9.8217
Interquartile range	5.9700	1.2900	14.6750	3.7400	1.4300	13.8900
Minimum	0.9400	1.6800	0.9400	1.0800	1.0800	7.8500
Maximum	82.2700	0.9400	82.2700	21.7400	5.4300	21.7400
Neutrophil gelatinase-associated lipocalin (ng/mL)						
Mean	50.9619	40.9860	57.1969	31.7993	29.4342	45.9900
Median	42.7450	38.5650	48.2750	33.9900 *	31.4500 *	45.9900
Standard deviation	33.2534	13.0503	40.4101	9.6271	8.1668	1.1455
Interquartile range	26.610	19.5500	45.1250	8.5000	6.9100	1.6200
Minimum	16.2600	27.3500	16.2600	12.1800	12.1800	45.1800
Maximum	183.860	68.620	183.860	46.800	40.290	46.8000
Transforming growth factor beta1 (ng/mL)						
Mean	0.5904	0.7370	0.4988	0.7129	0.7725	0.3550
Median	0.5500	0.5650	0.4800 *	0.4500 *	0.4500	0.3550
Standard deviation	0.3749	0.4874	0.2617	0.6867	0.7270	0.1344
Interquartile range	0.3600	0.7300	0.3600	0.8000	0.9850	0.1900
Minimum	0.1200	0.2000	0.1200	0.0400	0.0400	0.2600
Maximum	1.560	1.560	1.120	2.140	2.140	0.4500
Procalcitonin (ng/mL)						
Mean	521.4130 *	491.9510	539.8267	514.0858 *	494.1470	633.7185
Median	514.0541 *	460.8395	514.0541 *	514.0541 *	514.0541	633.7185
Standard deviation	107.4454	43.3750	131.0408	87.8889	25.6711	244.4879
Interquartile range	102.3431	53.2146	75.7362	53.2146	53.2146	345.7581
Minimum	402.1707	460.8395	402.1707	460.8395	460.8395	460.8395
Maximum	1001.430	460.8395	1001.430	806.598	514.054	806.5976
Cortisol (ng/mL)						
Mean	155.8308	108.6880	185.2950	114.9007	113.2742	124.6600
Median	124.3000	99.0450	162.5950	106.7000	106.7000	124.6600
Standard deviation	95.3656	37.5266	109.104	27.8167	25.6711	37.9716
Interquartile range	122.270	63.3100	160.225	47.8100	45.5250	57.3000
Minimum	64.1300	64.1300	67.6700	72.1500	72.1500	97.8100
Maximum	445.320	180.820	445.320	154.370	154.370	151.5100

Note: * statistically significant results (comparison of biomarker levels before and after surgery—Wilcoxon’s test).

**Table 4 biomedicines-11-01880-t004:** The descriptive characteristics of the studied biomarkers postoperatively.

Characteristics	Laparoscopy			Open Surgery		
	Total n = 26	No Inflammation n = 10	Inflammation n = 16	Total n = 14	No Inflammation n = 12	Inflammation n = 2
C-reactive protein (mg/L)						
Mean	63.1962	8.9800	97.0813	10.9714	7.3250	32.8500
Median	31.3000 *	8.1500 *	78.9000 *	3.4000 *	3.2000 *	32.8500
Standard deviation	78.2525	6.5571	83.6398	16.0484	10.6977	31.1834
Interquartile range	100	4.5000	92.7000	8.4000	7.4000	44.1000
Minimum	1.8000	1.9000	1.8000	0.5000	0.5000	10.8000
Maximum	272.700	23.600	272.700	54.900	38.700	54.9000
Interleukin-6 (pg/mL)						
Mean	23.8177	18.6800	27.0288	10.9786	11.3500	8.7500
Median	12.8200 *	5.4700	16.3050	7.8400 *	7.8350	8.7500
Standard deviation	31.5849	34.3939	30.4026	7.9414	8.4433	4.9073
Interquartile range	25.5000	12.1700	29.6100	6.0100	5.7250	6.9400
Minimum	2.8100	3.2000	2.8100	4.3500	4.3500	5.2800
Maximum	121.250	115.2900	121.2500	31.8200	31.8200	12.2200
Neutrophil gelatinase-associated lipocalin (ng/mL)						
Mean	54.8715	44.2330	61.5206	46.5771	38.2233	96.7000
Median	42.8400	38.0700	52.9500	39.2800 *	38.0200 *	96.7000
Standard deviation	29.5212	17.6653	33.8032	25.0879	14.3234	7.9479
Interquartile range	26.610	9.5400	49.9100	23.0900	17.7850	11.2400
Minimum	18.8900	31.3700	18.8900	14.7200	14.7200	91.0800
Maximum	128.270	87.410	128.270	102.320	69.070	102.3200
Transforming growth factor beta1 (ng/mL)						
Mean	0.6592	0.6050	0.6931	0.9257	0.9083	1.0300
Median	0.5500	0.4750	0.6000 *	0.8200 *	0.6450	1.0300
Standard deviation	0.4883	0.4344	0.5302	0.6809	0.7359	0.2121
Interquartile range	0.7800	0.7900	0.7900	0.9900	1.0850	0.3000
Minimum	0.1100	0.1400	0.1100	0.0900	0.0900	0.8800
Maximum	2.030	1.410	2.030	2.420	2.420	1.1800
Procalcitonin (ng/mL)						
Mean	820.0720 *	592.5129	962.2965	678.9371 *	683.5604	651.1977
Median	514.0541 *	514.0541	563.1826 *	514.0541 *	514.0541	651.1977
Standard deviation	687.2618	317.8954	818.8295	384.6610	410.0183	269.2072
Interquartile range	394.1524	53.2146	610.2199	345.758	289.2115	380.7164
Minimum	402.1707	402.1707	460.8395	460.8395	460.8395	460.8395
Maximum	3578.271	1480.581	3578.271	1912.364	1912.364	841.5559
Cortisol (ng/mL)						
Mean	166.2531	151.3150	175.5894	139.4650	115.6150	282.5650
Median	140.7850	132.0500	147.7450	120.4650	108.3550	282.5650
Standard deviation	109.6459	60.2585	132.7291	97.4022	63.8657	175.1716
Interquartile range	98.0600	93.7400	610.2199	84.8200	75.3000	247.7300
Minimum	5.5000	70.8300	5.5000	52.2300	52.2300	158.7000
Maximum	506.690	261.300	506.590	406.430	284.780	406.4300

Note: * statistically significant results (comparison of biomarker levels before and after surgery—Wilcoxon’s test).

**Table 5 biomedicines-11-01880-t005:** The correlations between postoperative CRP, NGAL and PCT concentrations and other biomarkers.

C-reactive protein (mg/L)		
	r–correlation coefficient	*p*-value
IL-6	0.5082	0.001 *
NGAL	0.3674	0.020 *
TGF-β1	0.1991	NS
PCT	0.6086	0.000 *
Cortisol	0.3032	NS
Neutrophil gelatinase-associated lipocalin (ng/mL)		
	r–correlation coefficient	*p*-value
IL-6	0.3820	0.015 *
TGF-β1	0.3355	0.034 *
PCT	0.2055	NS
Cortisol	0.2863	NS
Procalcitonin (ng/mL)		
	r–correlation coefficient	*p*-value
IL-6	0.3323	0.036 *
TGF-β1	0.2711	NS
Cortisol	0.3096	NS

Note: * statistical significance. NS, no statistical significance.

## Data Availability

The data presented in this study are available on request from the corresponding author. The data are not publicly available.

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
