# Peer review of "Post-Traumatic Stress Response and Appendicitis in Children—Clinical Usefulness of Selected Biomarkers"

_biomedicines, 2023, doi:10.3390/biomedicines11071880_

Round 1
Reviewer 1 Report
Reviewer’s comments to Author:
1. To determine the serum levels of the acute phase proteins, C-reactive protein (CRP), procalcitonin (PCT) and neutrophil gelatinase-as-sociated lipocalin (NGAL), interleukin 6 (IL-6), transforming growth factor-beta1 and cortisol (HC) of patients with Uncomplicated and Complicated appendicitis statistically significantly different?
2. According to the choice of the surgery method that patients Uncomplicated and Complicated appendicitis need to perform the operation, are there any criteria for the operation? Including clinical manifestations, laboratory data, and related imaging examinations, etc.
3. In addition to the lab data based on Pathophysiology logical inference, how to clearly define the significant increase in the concentration of Post-traumatic Stress Response of theses inflammatory biomarkers caused by Uncomplicated and Complicated Appendicitis after surgery is related to the surgical method? Is it related to the disease development process, severity, or is the source of the relevant bacterial or viral infection, or patient age distribution relevant?
4. According to the results of this study, the concentration of specific inflammatory biomarkers showed a significantly increase after surgery. Can these indicators provide any suggestions for pediatric surgeons in the future? Or use these indicators to make the choice of surgical method?
Reviewer 2 Report
Dear authors
This study has assessed post-traumatic inflammatory participants and their rate in pediatrics. The sections of the study have been described profoundly. This study is relatively novel and interesting findings have been demonstrated.
The conclusion section has been described sufficiently with good presentation.
The study details are interesting for readers.
Please describe what predisposes children to the post-traumatic inflammatory event.
Regards
Dear editor
The English writing of the manuscript is suitable.
